# Linear viscoelastic properties of the vertex model for epithelial tissues

**Sijie Tong[1], Navreeta K. Singh[1], Rastko Sknepnek[2,3]\*, Andrej Košmrlj[1,4]\***

**1** Department of Mechanical and Aerospace Engineering, Princeton University, Princeton, New Jersey, United States of America, **2** School of Science and Engineering, University of Dundee, Dundee, United Kingdom, **3** School of Life Sciences, University of Dundee, Dundee, United Kingdom, **4** Princeton Institute of Materials, Princeton University, Princeton, New Jersey, United States of America

\* r.sknepnek@dundee.ac.uk (RS); andrej@princeton.edu (AK)

## Abstract

Epithelial tissues act as barriers and, therefore, must repair themselves, respond to environmental changes and grow without compromising their integrity. Consequently, they exhibit complex viscoelastic rheological behavior where constituent cells actively tune their mechanical properties to change the overall response of the tissue, e.g., from solid-like to fluid-like. Mesoscopic mechanical properties of epithelia are commonly modeled with the vertex model. While previous studies have predominantly focused on the rheological properties of the vertex model at long time scales, we systematically studied the full dynamic range by applying small oscillatory shear and bulk deformations in both solid-like and fluid-like phases for regular hexagonal and disordered cell configurations. We found that the shear and bulk responses in the fluid and solid phases can be described by standard spring-dashpot viscoelastic models. Furthermore, the solid-fluid transition can be tuned by applying predeformation to the system. Our study provides insights into the mechanisms by which epithelia can regulate their rich rheological behavior.

**Data Availability Statement:** The simulation and analysis codes and initial configurations are available on GitHub: https://github.com/sknepneklab/RheoVM.

## Author summary

Epithelial tissues line organs and cavities in the body, and serve as barriers that separate organisms from their environment. Epithelia are robust yet adaptable; they have the ability to change their own viscoelastic behavior in response to internal or external stimuli by actively tuning the mechanical properties of the constituent cells and interactions between them. The mesoscopic mechanics of epithelia are commonly described with the vertex model. Here we present a detailed study of the linear rheological properties of the vertex model for both regular hexagonal and disordered cell configurations over a wide range of driving frequencies. The linear viscoelastic responses of the vertex model are mapped to standard spring-dashpot models. Our work, therefore, shows that the vertex model is a suitable base model to study the rich rheological behavior of epithelial tissues.

**Funding:** This research was primarily supported by NSF through the Princeton University's Materials Research Science and Engineering Center DMR-2011750 (to AK) and by the Project X Innovation Research Grant from the Princeton School of Engineering and Applied Science awarded to AK. This research was also supported by the UK Biotechnology and Biological Sciences Research Council (Award BB/N009789/1) awarded to RS. AK and RS initiated this project during the KITP program "Symmetry, Thermodynamics and Topology in Active Matter" (ACTIVE20), which was supported in part by the National Science Foundation under Grant No. NSF PHY-1748958. The funders had no role in study design, data collection and analysis, decision to publish, or preparation of the manuscript.

**Competing interests:** The authors have declared that no competing interests exist.

## Introduction

The development and maintenance of tissues requires close coordination of mechanical and biochemical signaling [1–3]. There is, for instance, mounting evidence for the key role played by tissue material properties and their regulation during embryonic development [4]. Tissues must be able to adjust their mechanical properties in response to internal and external stimuli. In particular, epithelial tissues, which line all cavities in the body and demarcate organs, must sustain substantial mechanical stresses while also supporting numerous biological processes such as selective diffusion and absorption/secretion of molecules [5]. In homeostasis, epithelia must maintain their shape and resist deformation while remaining flexible. The tissue must also be able to regenerate and repair itself, often with fast turnover, e.g., in gut epithelia [6]. Furthermore, in morphogenesis, the epithelial tissue must take up a specific shape and function [7], but this shape is lost during metastasis when cancer cells invade surrounding healthy tissues [8]. All of these processes require that cells be able to move, often over distances much larger than the cell size. During cell migration, however, the epithelial tissue must maintain its integrity. It is, therefore, not surprising that epithelia exhibit rich viscoelastic behavior [9]. Unlike passive viscoelastic materials, an epithelial tissue can actively tune its rheological response, making the study of its rheology not only important for understanding biological functions but also an interesting problem from the perspective of the physics of active matter systems [10].

Collective cell migration has been extensively studied in biology [11] and biophysics [12]. In vitro studies of confluent cell monolayers [13–17] focused on the physical aspects of force generation and transmission and showed that cell migration is an inherently collective phenomenon. Some aspects of collective cell migration are remarkably similar to the slow dynamics of structural glasses [18–24]. This suggests that many of the observed behaviors share common underlying mechanisms and can be understood, at least at mesoscales (i.e., distances beyond several cell diameters), using physics of dense active systems [25]. A particularly intriguing observation is that tuning cell density [18, 26, 27], strength of cell-cell and cell-substrate interactions [28], or cell shape parameters [21, 29] can stop collective migration. In other words, the epithelium undergoes a fluid to solid transition. Signatures of such behavior have been reported in several in vitro [19, 30] and developmental systems [31–33]. This suggests that important aspects of morphogenetic development may rely on epithelial tissue's ability to undergo phase transitions [4].

How an epithelial tissue responds to external and internal mechanical stresses depends on its rheological (i.e., material) properties. While there have been numerous studies focusing on the rheology of a single cell [34–36], much less is known about tissue rheology, particularly during development. In order to develop a comprehensive understanding of epithelial tissue mechanics, such insight is key. Though single cell measurements are valuable, the mechanics of an epithelial tissue can be drastically different from that of its constituent cells. The stiffness of cell monolayers, for example, is orders of magnitude higher than the stiffness of constituent cells, while the time dependent mechanical behaviors of monolayers in response to deformation vary depending on the magnitude of loading [37]. Embryonic cell aggregates have been shown to behave elastically (i.e., solid-like) at short timescales, but they flow like fluids at long timescales, which facilitates both the robustness needed to maintain integrity and the flexibility to morph during development [9]. Experiments have characterized the mechanical behaviors of epithelial tissues at various loading conditions, which led to a phenomenological description that models the relaxation properties of epithelial monolayers based on fractional calculus [38]. Notably, a recent particle-based model that includes cell division and apoptosis provided a plausible microscopic model for nonlinear rheological response [39]. Particle-based models

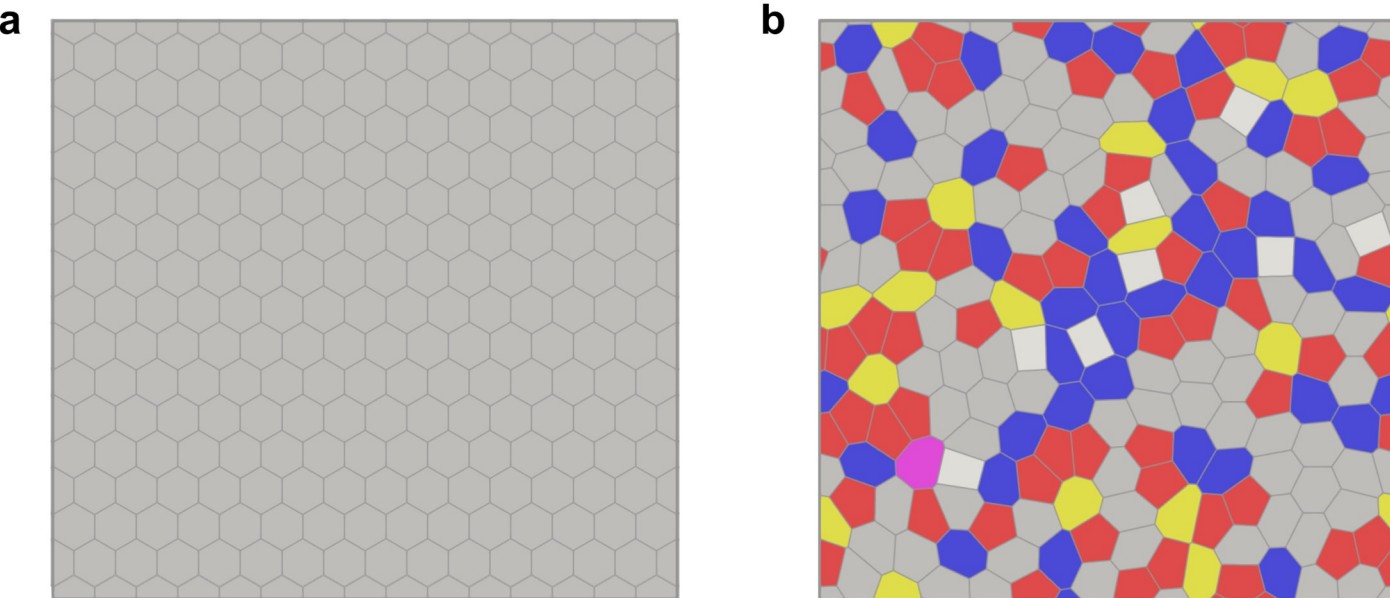

**Fig 1. Epithelial tissue is represented as a polygon tiling of the plane subject to periodic boundary conditions.** We studied the rheology of the vertex model for both (a) regular hexagonal and (b) disordered tilings. Colors represent the number of neighbors of each cell; 4-white, 5-red, 6-gray, 7-blue, 8-yellow.

are, however, unable to capture geometric aspects such as cell shape. It is, therefore, necessary to investigate rheological response in geometric models.

The vertex model [40–42] and more recent, closely related Voronoi models [21, 43, 44] have played an important role in modeling mechanics of epithelial tissues since they account for the shapes of individual cells and provide a link to cellular processes, such as cell-cell adhesion, cell motility, and mitosis [42]. These geometric models are also able to capture the solid to fluid transition and demonstrate rich and unusual nonlinear mechanical behavior [45, 46]. While the mechanical properties of the vertex and Voronoi models have been extensively studied, most works to date focused on the long-time behavior. These include studies of the quasistatic shear modulus [20], effective diffusion constant of cells related to the tissue viscosity [21], correlations between a structural property called "softness" and the likelihood of topological rearrangements of cells [47], and steady state flow profiles around a sphere dragged through the tissue [48]. The rheological properties of the vertex model that cover a broad range of timescales, however, have not yet been systematically explored. In this paper, we model the response of a model epithelial tissue adhered to a substrate by studying the response of the regular hexagonal and disordered cell configurations in the vertex model (see Fig 1) to applied oscillatory shear and bulk deformations of small amplitude, i.e., in the linear response regime. We measured the response stresses and used them to compute the storage and loss moduli in both the solid and fluid phases. We show that the dynamical response of the vertex model can be fitted to standard spring-dashpot viscoelastic models over seven decades in the driving frequency and that the solid-fluid transition can be tuned by applying pre-deformation to the system. Thus we argue that the vertex model makes a suitable basis for studies of dynamics of epithelial tissues beyond the quasistatic limit.

## Model and methods

### Vertex model

In the vertex model, the state of an epithelial tissue is approximated as a polygonal tiling of the plane (Fig 1). The degrees of freedom are vertices, i.e., meeting points of three or more cell-cell

junctions. In the simplest formulation, junctions are assumed to be straight lines. The energy of the vertex model is a quadratic function of cell areas and perimeters [41], i.e.,

$$E = \sum_C \left[ \frac{K_C}{2} (A_C - A_{C0})^2 + \frac{\Gamma_C}{2} (P_C - P_{C0})^2 \right], \tag{1}$$

where $K_C$ and $\Gamma_C$ are the area and perimeter elastic moduli, and $A_C$ and $A_{C0}$ are the actual and preferred areas of cell $C$, respectively. Similarly, $P_C$ and $P_{C0}$ are the actual and preferred perimeters of the same cell. In this work, we assumed $K_C$, $\Gamma_C$, $A_{C0}$, and $P_{C0}$ to be identical for all cells (i.e., $K_C \equiv K$, $\Gamma_C \equiv \Gamma$, $A_{C0} \equiv A_0$, $P_{C0} \equiv P_0$). Further, we fixed the values of $K$ and $A_0$, and measured the energy in units of $KA_0^2$, stresses in units of $KA_0$, and lengths in units of $A_0^{1/2}$. Since the ratio between the perimeter and area elastic moduli does not qualitatively change the behavior of the vertex model [20, 41], we fixed that ratio to $\Gamma/(KA_0) \approx 0.289$ for all simulations. The only variable parameter in simulations was the preferred cell perimeter $P_0$, which sets the dimensionless cell-shape parameter, defined as the ratio $p_0 = P_0/\sqrt{A_0}$.

The cell-shape parameter, $p_0$, plays a central role in determining whether the system behaves as a fluid or solid [20]. Bi, *et al.* [20] argued that the rigidity transition occurs at $p_0 = p_c \approx 3.812$ for a disordered polygonal tiling, while Merkel, *et al.* [49] reported $p_0 = p_c \approx 3.92$. For a regular hexagonal tiling, the transition point is at $p_c = \sqrt{8\sqrt{3}} \approx 3.722$ [50]. In the fluid phase, the energy barrier for neighbor exchanges vanishes and cells can flow past each other [51]. As $p_0$ is reduced below $p_c$, the energy barrier becomes finite, neighbor exchanges cease and the system becomes solid. While the transition point for regular hexagonal tilings can be understood in terms of the mechanical stability and the excess perimeter [45, 52], the mechanism that leads to a larger value for random tilings is more subtle and only partly understood [53]. For example, recent studies [49, 54] have shown that the rigidity transition of random tilings depends on the procedure used to generate the tilings. The presence of vertices with coordination greater or equal to four, as well as the presence of cells with five or less neighbors, increases the critical value of the cell-shape parameter $p_0$ [54].

## Simulation setup

We first studied the rheology of regular hexagonal tilings (Fig 1a) subject to periodic boundary conditions. The shape of the simulation box was chosen to be as close to a square as allowed by the geometry of a hexagon, and the area of the box was such that it accommodated $N$ cells of area $A_C$ that matched the preferred areas $A_0$. Most simulations started with hexagonal tiling with $N_x = 15$ cells in the horizontal direction (i.e., $N = 240$ cells in total, Fig 1a). Simulations of larger system sizes ($N_x = 37$, 51, i.e., $N = 1406$, 2652 total cells, respectively) were performed for a subset of values of $p_0$ to explore the finite size effects. No quantitative differences between the system with $N = 240$ cells and larger systems were observed. All simulations were performed with an in-house code [55] and snapshots of cell configurations were visualized with ParaView [56].

For the solid phase with $p_0 \lesssim 3.722$, the ground state of the energy in Eq (1) is the honeycomb lattice [50], and it was directly used to investigate rheological properties. Note that there was some residual hydrostatic stress due to the mismatch of actual cell perimeters $P_C$ from their preferred values $P_0$, which could be eliminated by appropriate rescaling of the simulation box. This hydrostatic stress, however, does not qualitatively affect the rheological behavior of the system (see Sec D in S1 File for further discussion). For the fluid phase with $p_0 \gtrsim 3.722$, the hexagonal tiling corresponds to a saddle point of the energy in Eq (1) [50]. A small random perturbation was applied to each vertex, i.e., each vertex was displaced from its original

position in the hexagonal tiling by a vector $\delta\mathbf{r}_i = \delta x_i \mathbf{e}_x + \delta y_i \mathbf{e}_y$, where $\delta x_i$ and $\delta y_i$ were Gaussian random variables with zero mean and standard deviation $1.5 \times 10^{-4}\sqrt{A_0}$; the system was then relaxed using the FIRE algorithm [57] to reach a local energy minimum with the relative accuracy of $10^{-12}$. Note that the energy landscape in the fluid phase has many local minima and a large number of soft modes (see Sec H in S1 File). We repeated simulations to investigate rheological properties for multiple configurations corresponding to different local energy minima.

The study of the rheological properties of the vertex model for a regular hexagonal tiling is appealing since one can make comparisons to analytical treatments. The regular hexagonal tiling is, however, a rather crude approximation of real epithelial tissues, which are typically irregular [58]. Therefore, we also investigated the rheology of the vertex model of disordered tilings (Fig 1b). The disordered tilings were created as follows (see Fig A in the Sec A in S1 File for a schematic illustration). We used the random sequential addition algorithm [59] to place $N = 200$ seed points inside a square box of size $L = 15$ without overlaps. We then created periodic images of the seed points and used SciPy to build the periodic Voronoi tessellation [60]. The preferred area of each cell was set to $A_0 = L^2/N$. The energy of the system given in Eq (1) was then relaxed using the FIRE algorithm to reach a local minimum. During the energy minimization, T1 transitions (exchanges of cell neighbors) were allowed but were not common. We generated an ensemble of 10 different random initial configurations using different values of the random number generator seed and repeated rheology simulations for each of those configurations to probe the rheology for a range of values of $p_0$.

## Dynamics and probing the rheology

In order to probe the dynamic response of the vertex model, we need to specify the microscopic equations of motion for vertices. Assuming the low Reynolds number limit, which is applicable to most cellular systems due to their slow speed, inertial effects can be neglected [61]. The equations of motion are then a force balance between friction with the substrate and elastic forces $\mathbf{F}_i$ due to deformations of cell shapes, i.e.,

$$\gamma\dot{\mathbf{r}}_i = \mathbf{F}_i. \tag{2}$$

Here, we assume that friction between the tissue and the substrate arises from binding and unbinding of adhesion molecules. In particular, on time scales much longer than the characteristic unbinding time, the tissue–substrate adhesive bonds undergo stick-and-slip processes leading to a form of viscous friction [62–64]. In the above Eq (2), $\mathbf{r}_i$ is the position vector of vertex $i$ in a laboratory frame of reference, $\mathbf{F}_i = -\nabla_{\mathbf{r}_i}E$ is the mechanical force on vertex $i$ due to deformation of cells surrounding it, $\gamma$ is the friction coefficient, and dot denotes the time derivative. Therefore, each vertex experiences dissipative drag proportional to its instantaneous velocity. We fixed the value of $\gamma$ in simulations, which sets the unit of time as $\gamma/(KA_0)$. Furthermore, we neglected thermal fluctuations and hence omit the stochastic term in Eq (2). This is a reasonable assumption since typical energy scales in tissues significantly exceed the thermal energy, $k_BT$, at room temperature $T$, where $k_B$ is the Boltzmann constant. It is, however, worth noting that there are other sources of stochasticity in epithelia (e.g., fluctuations of the number of force-generating molecular motors) which are important for tissue scale behaviors [65]. Here, we did not consider such effects but note that they could be directly included in the model as additional forces in Eq (2).

We applied an oscillatory affine deformation to investigate the rheological behavior of the vertex model. The affine deformation can be described by a deformation gradient tensor defined as $\hat{\boldsymbol{F}} = \partial\mathbf{x}/\partial\mathbf{X}_0$, where the mapping $\mathbf{x} = \mathbf{x}(\mathbf{X}_0, t)$ maps the reference configuration $\mathbf{X}_0$

to a spatial configuration **x** at time $t$. Note that the affine deformation was applied to all vertices inside the bounding box as well as to the periodic images of vertices, which ensured deformation of cells across periodic boundaries. Specifically, for a vertex $i$ with a position vector $\mathbf{r}_i(t)$ in a deformed configuration, we also consider its periodic images at positions $\mathbf{r}_i(t) + m_1\mathbf{a}_1(t) + m_2\mathbf{a}_2(t)$, where $\mathbf{a}_{1,2}(t)$ are the unit cell vectors of the periodic box and $m_{1,2}$ are integers. For cells that cross the periodic boundary, junction lengths are computed using the minimum image convention. Note that the unit cell vectors of the periodic box transform as $\mathbf{a}_{1,2}(t) = \hat{F}(t)\mathbf{a}_{1,2}(0)$ due to the affine deformation.

The deformation gradient of simple shear is $\hat{F}(t) = \left( \begin{smallmatrix} 1 & \epsilon(t) \\ 0 & 1 \end{smallmatrix} \right)$ and of biaxial deformation is $\hat{F}(t) = \left( \begin{smallmatrix} 1+\epsilon(t) & 0 \\ 0 & 1+\epsilon(t) \end{smallmatrix} \right)$, where $\epsilon(t) = \epsilon_0 \sin(\omega_0 t)$ and $\omega_0$ is the frequency of the oscillatory deformation. In all simulations, we used a small magnitude of deformation, i.e., $\epsilon_0 = 10^{-7}$, so that we probed the linear response and the measured moduli were independent of the magnitude of the deformation. In every time step after the affine deformation was applied, the system evolved according to the overdamped dynamics in Eq (2). During the oscillatory deformations, T1 transitions were allowed but were not common. Equations of motion were integrated using the first-order Euler method [66] with the time step $\Delta t \approx 0.00866\gamma/(KA_0)$ when the frequency of oscillatory deformation $\omega_0\gamma/(KA_0) < 29.02$, but with a smaller time step $\Delta t \approx 0.000866\gamma/(KA_0)$ when $\omega_0\gamma/(KA_0) > 29.02$ so that there were enough sampling points (at least 25) over one period of oscillatory deformation.

The response stress tensor, $\hat{\boldsymbol{\sigma}}_C(t)$, for each cell $C$ was computed using the formalism introduced in Refs. [67–69] as

$$\hat{\boldsymbol{\sigma}}_C = -\Pi_C\hat{I} + \frac{1}{2A_C}\sum_{e\in C}\mathbf{T}_e \otimes \mathbf{l}_e, \tag{3}$$

where the summation is over all junctions $e$ belonging to cell $C$. Here, $\Pi_C = -\frac{\partial E}{\partial A_C} = -K(A_C - A_0)$ is the hydrostatic pressure inside a cell, $\hat{I}$ is the unit tensor, and $\mathbf{T}_e = \frac{\partial E}{\partial \mathbf{l}_e} = \Gamma(P_C - P_0)\mathbf{l}_e/|\mathbf{l}_e|$ is the tension along the junction $e$ with $\mathbf{l}_e$ being a vector joining the two vertices on it [67–69]. The average stress tensor $\hat{\boldsymbol{\sigma}}(t) = \sum_C w_C\hat{\boldsymbol{\sigma}}_C(t)$, with $w_C = A_C/\Sigma_C A_C$, was used as a measure for the response of the system. Measurements of the response stresses for each cell [see Eq (3)] and the entire system were taken 25 times within each cycle of oscillatory deformation.

To ensure that we were probing the steady state, we performed the following analysis. For example, in the case of shear deformation, the shear stress signal $\tau(t) = \hat{\sigma}_{xy}(t)$ was divided into blocks of length $T = 3T_0$, each containing 3 cycles of the time period $T_0 = 2\pi/\omega_0$ of the driving shear deformation. Within each block $n$, we performed the Fourier transform of $\tau(t)$ and obtained $\tilde{\tau}_n(\omega)$ as

$$\tilde{\tau}_n(\omega) = \frac{1}{T}\int_{(n-1)T}^{nT} \tau(t)e^{i\omega t}dt, \tag{4}$$

where $n$ is a positive integer. Similar Fourier transform analysis was performed for the strain, $\epsilon(t)$, of which the Fourier transform is denoted as $\tilde{\epsilon}(\omega)$. The length of the simulation was chosen such that it contained a sufficient number of blocks in order for the $\tilde{\tau}_n(\omega_0)$ to reach a steady state value $\tilde{\tau}(\omega_0)$. The obtained steady state value of $\tilde{\tau}(\omega_0)$ was used to calculate the dynamic shear modulus $G^*(\omega_0) = \tilde{\tau}(\omega_0)/\tilde{\epsilon}(\omega_0)$ at a given frequency $\omega_0$ of applied shear strain. We ensured that simulations ran long enough to reach a steady state. An analogous procedure

 

was applied to the hydrostatic stress, $\sigma(t) = \frac{1}{2}[\hat{\sigma}_{xx}(t) + \hat{\sigma}_{yy}(t)]$, in the case of the bulk deformation. Please refer to Sec C in S1 File for a representative example of the steady state analysis.

## Results

### Response to a shear deformation

The hexagonal ground state in the solid phase and states corresponding to local energy minima in the fluid phase were used to investigate the rheological behavior by applying an oscillatory affine shear deformation to the substrate (Fig 2a and 2b). Due to the binding and unbinding of adhesive molecules, deformation of the tissue follows the deformation of the substrate on short timescales, and then tissue can relax on longer timescales. Thus, at each time step, we first applied the affine shear deformation to the simulation box and all vertices, which was followed by internal relaxation of the vertices according to Eq (2). The affine simple shear deformation can be described by a deformation gradient tensor, $\hat{F} = \begin{pmatrix} 1 & \epsilon(t) \\ 0 & 1 \end{pmatrix}$, where $\epsilon(t) = \epsilon_0 \sin(\omega_0 t)$. Sufficiently small amplitude $\epsilon_0 = 10^{-7} \ll 1$ was used to probe the linear response properties.

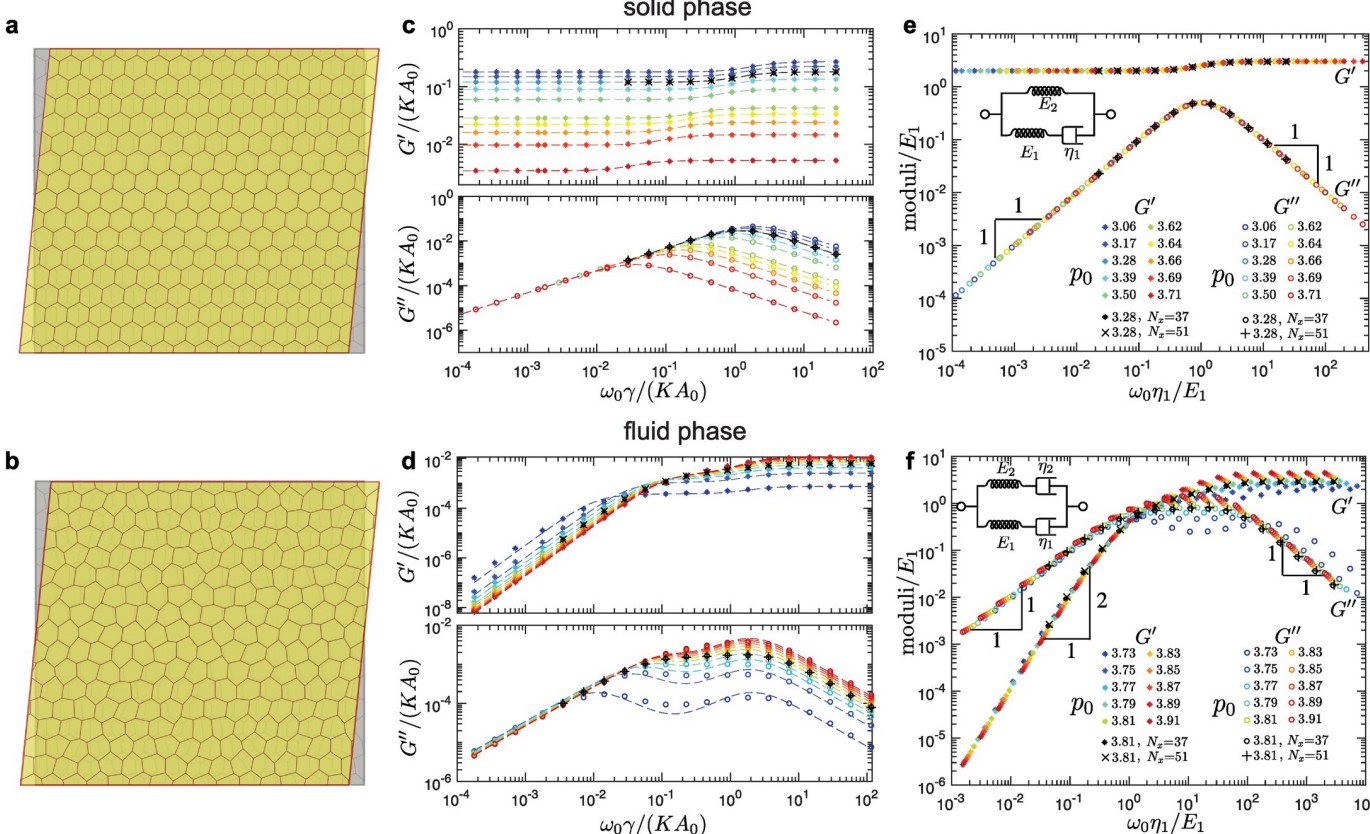

**Fig 2. Storage and loss shear moduli in the solid (top row) and fluid phase (bottom row) for hexagonal tilings.** (a-b) An overlay of the representative reference (grey) and sheared (yellow) configurations in (a) the solid and (b) the fluid phase. The magnitude of the shear is highly exaggerated for demonstration purposes. (c-d) Representative storage ($G'$) and loss ($G''$) shear moduli as functions of the shearing frequency, $\omega_0$, for different values of the cell-shape parameter, $p_0$. Dashed curves are the fits based on (c) the Standard Linear Solid (SLS) model in the solid phase [see Eq (5)] and (d) the Burgers model in the fluid phase [see Eq (8)]. (e-f) The collapse of the moduli curves for different values of $p_0$ for (e) the solid phase and (f) the fluid phase. The insets show the representation of (e) the SLS model and (f) the Burgers model in terms of the springs and dashpots. The majority of the data corresponds to the system of nearly square shape with $N_x = 15$ cells in the horizontal direction, and we also show examples of larger systems with $N_x = 37$ and $N_x = 51$ cells in the horizontal direction.

We measured the response stresses as described in the Model and methods section above. The dynamic shear modulus $G^*(\omega_0) = \tilde{\tau}(\omega_0)/\tilde{\epsilon}(\omega_0)$ was then calculated at a given frequency $\omega_0$ of applied shear strain, where $\tilde{\tau}(\omega)$ and $\tilde{\epsilon}(\omega)$ are the Fourier transforms of the response shear stress $\tau(t) = \hat{\sigma}_{xy}(t)$ and the applied strain $\epsilon(t)$, respectively (see Model and methods). We ensured that simulations ran long enough to reach a steady state (see Model and methods and Sec C in S1 File). The real part of the dynamic shear modulus, $G' = \text{Re}(G^*)$, is the storage shear modulus and the imaginary part, $G'' = \text{Im}(G^*)$, is the loss shear modulus. The storage shear modulus corresponds to the in-phase response and measures the elastic (i.e., reversible) response of the system, while the loss shear modulus corresponds to the out-of-phase response and measures the system's irreversible dissipation [70] (see also Sec B in S1 File). For systems under an oscillatory simple shear, storage and loss shear moduli were obtained for different values of $p_0$ and different system sizes in the solid and the fluid phases for a broad range of driving frequencies $\omega_0$ spanning over seven orders of magnitude, as shown in Fig 2c and 2d. Most simulations were performed for systems with nearly square shapes with $N_x = 15$ cells in the horizontal direction. We repeated several simulations for systems with $N_x = 37$ and $N_x = 51$, which showed that the finite size effects are negligible (Fig 2c–2f).

In the solid phase there are two different regimes (see Fig 2c). At low frequencies, $\omega_0$, the storage shear modulus $G'$ has a constant value, while the loss shear modulus scales as $G'' \propto \omega_0$. At high frequencies, the storage shear modulus $G'$ has a higher constant value, while the loss shear modulus scales as $G'' \propto \omega_0^{-1}$. Such rheological behavior is characteristic of the Standard Linear Solid (SLS) model [70]. Storage and loss shear moduli for the SLS model are [70], respectively,

$$G'_{\text{SLS}}(\omega_0) \quad = \frac{E_2 + \dfrac{\eta_1^2}{E_1^2}\omega_0^2(E_1 + E_2)}{1 + \dfrac{\eta_1^2}{E_1^2}\omega_0^2}, \tag{5a}$$

$$G''_{\text{SLS}}(\omega_0) \quad = \frac{\omega_0\eta_1}{1 + \dfrac{\eta_1^2}{E_1^2}\omega_0^2}, \tag{5b}$$

where we used the representation of the SLS model (Fig 2e, inset) that consists of a spring with elastic constant $E_2$ connected in parallel with a Maxwell element, which comprises a spring with elastic constant $E_1$ and a dashpot with viscosity $\eta_1$ connected in series. The above expressions in Eq (5) were used to fit the storage and loss shear moduli obtained from simulations. The fitted curves, represented with dashed lines in Fig 2c, show an excellent match with the simulation data, indicating that the SLS model is indeed appropriate to describe the shear rheology in the solid phase. This was also confirmed in Fig 2e, where we collapsed the storage and loss shear moduli for different values of the shape parameter, $p_0$, by rescaling the moduli and frequencies with the fitted values of spring and dashpot constants. Note that the SLS response in the solid phase is consistent with recent experiments on suspended MDCK monolayers [71].

As the value of the $p_0$ increases, we observe that the storage shear modulus reduces at all frequencies and that the loss shear modulus reduces at high frequencies. Furthermore the crossover between the two regimes shifts towards lower frequencies (Fig 2c). This is because the elastic constants $E_1$ and $E_2$ decrease linearly with increasing $p_0$ and they become zero exactly at the solid-fluid transition with $p_0 = p_c \approx 3.722$ (Fig 3a). The dashpot constant $\eta_1$ is nearly independent of $p_0$ (Fig 3b) and scales with the friction parameter $\gamma$, which is the only source of dissipation in the vertex model. The crossover between the two regimes for both the storage and

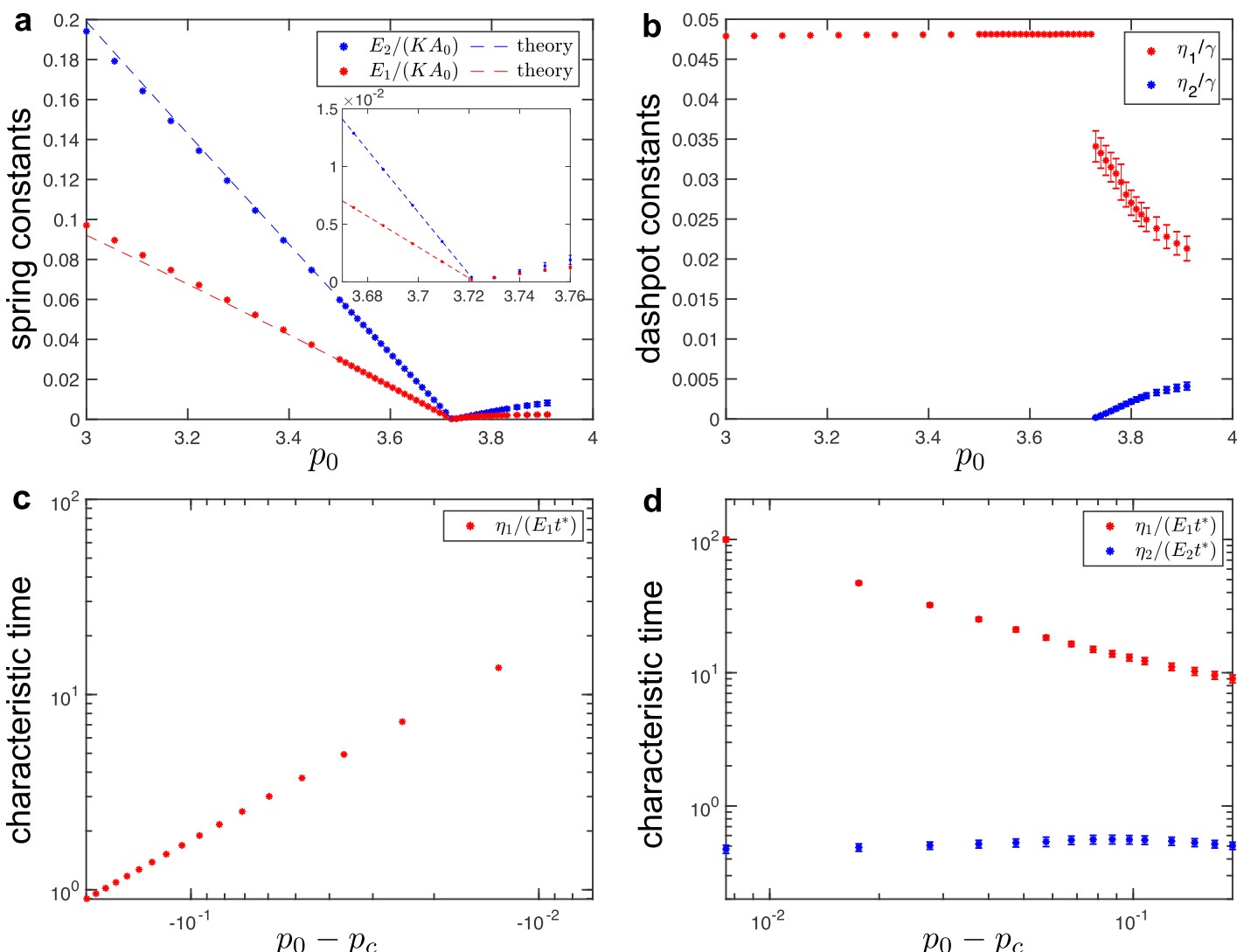

**Fig 3.** (a-b) Fitted values of spring-dashpot models for hexagonal tilings under simple shear. (a) Elastic constants as a function of target cell-shape parameter, $p_0$. In the solid phase (i.e., for $p_0 < p_c \approx 3.722$), fitted values of the spring constants show excellent match with the analytical predictions obtained from Eqs (6) and (7) (dashed lines). Inset shows the spring constants near the critical point. (b) Dashpot viscosity constants as a function of the target cell-shape parameter, $p_0$. (c-d) Characteristic timescales in (c) the solid and (d) fluid phase for hexagonal tilings obtained from the fitted values of the elastic constant and the dashpot viscosity. The normalization factor $t^* = \gamma/(KA_0)$ sets the unit of time. For the fluid phase (i.e., for $p_0 > p_c \approx 3.722$), errorbars correspond to the standard deviation for simulations that were repeated for configurations that correspond to different local energy minima.

loss shear moduli corresponds to a characteristic timescale, $\eta_1/E_1$, which diverges as $\sim \gamma(KA_0)^{-1}(p_c - p_0)^{-1}$ as $p_0$ approaches the solid-fluid transition (Fig 3c) due to the vanishing elastic constant (Fig 3a). Note that the values of the elastic constants $E_1$ and $E_2$ can be estimated analytically. In the quasistatic limit ($\omega_0 \to 0$), the external driving is sufficiently slow that the system can relax internally. In this limit, Murisic, *et al.* [72] showed that the storage shear modulus is

$$G'(\omega_0 \to 0) = E_2 = \frac{1}{2}KA_0\big(1 - [\alpha(p_0, \Gamma/KA_0)]^2\big), \qquad (6)$$

where $\alpha(p_0, \Gamma/KA_0)$ is a scaling factor chosen such that the hydrostatic stress vanishes once the system box size is rescaled from $L$ to $\alpha L$ (see Sec D in S1 File). In the high frequency limit

($\omega_0 \to \infty$), on the other hand, the system follows the externally imposed affine deformation and has no time for internal relaxation. Thus, by considering the energy cost for a hexagonal tiling under affine deformation, we obtained the storage shear modulus (see Sec G in S1 File)

$$G'(\omega_0 \to \infty) = E_1 + E_2 = 3\sqrt{3}\Gamma\left(1 - \frac{p_0}{p_c}\right). \tag{7}$$

The above Eqs (6) and (7) were used to extract the values of elastic constants $E_1$ and $E_2$, which showed excellent agreement with the fitted values from simulations (Fig 3a).

In the fluid phase, the storage and loss shear moduli show a markedly different behavior (Fig 2d). There are three different regimes with two crossover frequencies, which correspond to two characteristic timescales. At low frequencies, $\omega_0$, the storage shear modulus $G' \propto \omega_0^2$ and the loss shear modulus $G''(\omega_0) \propto \omega_0$. The storage modulus approaches 0 for $\omega_0 \to 0$, which indicates that the system is indeed a fluid. At high frequencies the storage shear modulus has a constant value, while the loss shear modulus scales as $G''(\omega_0) \propto \omega_0^{-1}$. To capture this behavior we used the Burgers model, which consists of two Maxwell models connected in parallel (Fig 2f, inset), to fit the shear moduli measured in the simulations. The storage and loss shear moduli for a Burgers model are [70], respectively,

$$G'_{\text{Burg}}(\omega_0) \quad = \frac{p_1 q_1 \omega_0^2 - q_2 \omega_0^2 (1 - p_2 \omega_0^2)}{p_1^2 \omega_0^2 + (1 - p_2 \omega_0^2)^2}, \tag{8a}$$

$$G''_{\text{Burg}}(\omega_0) \quad = \frac{p_1 q_2 \omega_0^3 + q_1 \omega_0 (1 - p_2 \omega_0^2)}{p_1^2 \omega_0^2 + (1 - p_2 \omega_0^2)^2}, \tag{8b}$$

where $p_1 = \eta_1/E_1 + \eta_2/E_2$, $p_2 = \eta_1 \eta_2/(E_1 E_2)$, $q_1 = \eta_1 + \eta_2$, $q_2 = \eta_1 \eta_2 (E_1 + E_2)/(E_1 E_2)$. The dashed curves in Fig 2d show fits of the storage and loss shear moduli for a range of values of $p_0$, which show good agreement with simulations. Unlike for the solid phase, it is not possible to collapse the data for storage and loss shear moduli onto single universal curves because the fluid phase is characterized by two independent timescales $\eta_1/E_1$ and $\eta_2/E_2$. Thus we show two different collapses for the storage and loss shear moduli in the low frequency range (Fig 2f) and in the high frequency range (Fig E in the Sec E in S1 File).

As the value of the $p_0$ decreases, we observe that both the storage and loss shear moduli reduce at intermediate and high frequencies, but they increase at low frequencies (Fig 2d). We also observe that the first crossover shifts towards lower frequencies, while the second crossover remains at approximately the same frequency. This is because the elastic constants $E_1$ and $E_2$ decrease linearly toward zero as $p_0$ approaches the solid-fluid transition at $p_c \approx 3.722$ (Fig 3a). The dashpot constant $\eta_2$ also decreases linearly toward zero, while the dashpot constant $\eta_1$ increases but remains finite as $p_0$ approaches the solid-fluid transition (Fig 3b). As a consequence, one of the characteristic timescales $\eta_1/E_1 \sim \gamma(KA_0)^{-1}(p_0 - p_c)^{-1}$ diverges, while the second timescale $\eta_2/E_2 \sim \gamma(KA_0)^{-1}$ remains finite as $p_0$ approaches the solid-fluid transition (Fig 3d). The diverging characteristic timescale captures the macroscopic behavior of the system, while the second timescale captures the microscopic details of the vertex model. Note that at the solid-fluid transition there is a discontinuous jump in the values of the dashpot constant $\eta_1$ (see Fig 3b). This is because at $p_0 = p_c$ the storage and loss shear moduli are identically equal to zero ($G'(\omega_0) = G''(\omega_0) \equiv 0$) due to the vanishing elastic constants ($E_1 = E_2 = 0$), while the dashpot constants can have arbitrary values [see Eqs (5) and (8)]. Finally, we note that the values of the spring and dashpot constants are somewhat sensitive to the local energy minimum configuration used to probe the response in the fluid phase. The errorbars in Fig 3 show standard deviation for different configurations that were obtained by using the same magnitude of the initial perturbation (see Model and methods). In Fig F in the Sec F in S1 File, we show how

the values of the spring and dashpot constants are affected when configurations were obtained by using different magnitudes of the initial perturbation.

Cells in real epithelial tissues are, however, unlikely to have a perfect hexagonal shape and form a honeycomb tiling. The observed tilings are disordered, often with a rather specific distribution of the number of neighbor cells conserved across several species [73]. To mimic the geometry of real tissues, we constructed an ensemble of uncorrelated disordered tilings of polygons corresponding to local energy minima at different values of $p_0$ (see Fig 1b and Model and methods). We then probed the shear rheology of each such configuration following the same procedure as for the hexagonal tilings. Fig 4 shows the average storage and shear moduli. We found that the critical value of $p_0$ for the solid-fluid transition was at $p_c \approx 3.93$, which is consistent with refs. [49, 54], but somewhat higher than what was reported in [20].

For $p_0 < p_c$ values corresponding to the system being deep in the solid phase (Fig 4a), the storage and loss moduli are described accurately by the Standard Linear Solid (SLS) model, the same as for the hexagonal tiling in the solid phase (Fig 2c). As $p_0$ increases, however, a second shoulder develops in the loss moduli (see $p_0 = 3.71$ and $p_0 = 3.77$ curves in Fig 4a), which indicates the presence of multiple time scales. The fits to the SLS model shown with the dashed lines also begin to deviate from the measured moduli. The scaling collapse is only possible for values of $p_0$ deep in the solid phase (Fig 4b). This supports the observation that SLS is no longer able to capture the rheology in the solid phase as $p_0$ approaches the critical point.

In the opposite limit, i.e., when the value of $p_0 > p_c$ is deep in the fluid phase (Fig 4c), the storage and loss moduli can be modeled with the Burgers model, the same as for the local energy minima states relaxed from the hexagonal tiling in the fluid phase (Fig 2d). The fits represented by the dashed lines, however, deviate from the measured moduli as $p_0$ decreases (see $p_0 = 3.97$ and $p_0 = 3.99$ curves in Fig 4c). Fig 4d shows the rescaling of the moduli and frequencies by the fitted spring and dashpot constants.

Near the critical value (i.e., for $p_0 = 3.93$ and $p_0 = 3.95$), the ensemble of random tilings contains the solid and the fluid configurations (see Fig I in the Sec I in S1 File), which was determined based on the presence or absence of non-trivial zero modes. We separated the solid and the fluid configurations and calculated average storage and loss shear moduli on each set. Fig 4e and 4f show the average storage and loss moduli for values of $p_0$ close to the critical value in the solid phase (Fig 4e) and the fluid phase (Fig 4f). The dashed curves are the fits based on the SLS model in the solid phase and the Burgers model in the fluid phase, which do not fully capture the behavior of the measured moduli curves due to the presence of multiple time scales. As the value of $p_0$ approaches the critical value, the spread of the moduli increases, especially for low frequencies, which is captured by the size of error bars. This can also be seen in Fig I in the Sec I in S1 File, which presents the raw data of storage and loss shear moduli at different values of $p_0$.

In Fig 5, we summarize the fitted values of spring-dashpot models. The values of spring constants decrease as the system approaches the solid-fluid transition, while the values of dashpot constants diverge near the transition. In the intermediate regime (shaded regions in Fig 5), the SLS model (in the solid phase) and the Burgers model (in the fluid phase) cannot accurately fit the measured moduli due to the presence of additional timescales.

## Response to bulk deformations

We further studied the bulk rheological properties of the hexagonal tilings by applying an oscillatory biaxial deformation to the substrate (Fig 6a and 6b) described by the deformation gradient $\hat{F} = \begin{pmatrix} 1+\epsilon(t) & 0 \\ 0 & 1+\epsilon(t) \end{pmatrix}$, where $\epsilon(t) = \epsilon_0 \sin(\omega_0 t)$. We applied a sufficiently small amplitude $\epsilon_0 = 10^{-7} \ll 1$ to probe the linear response properties characterized by the average normal

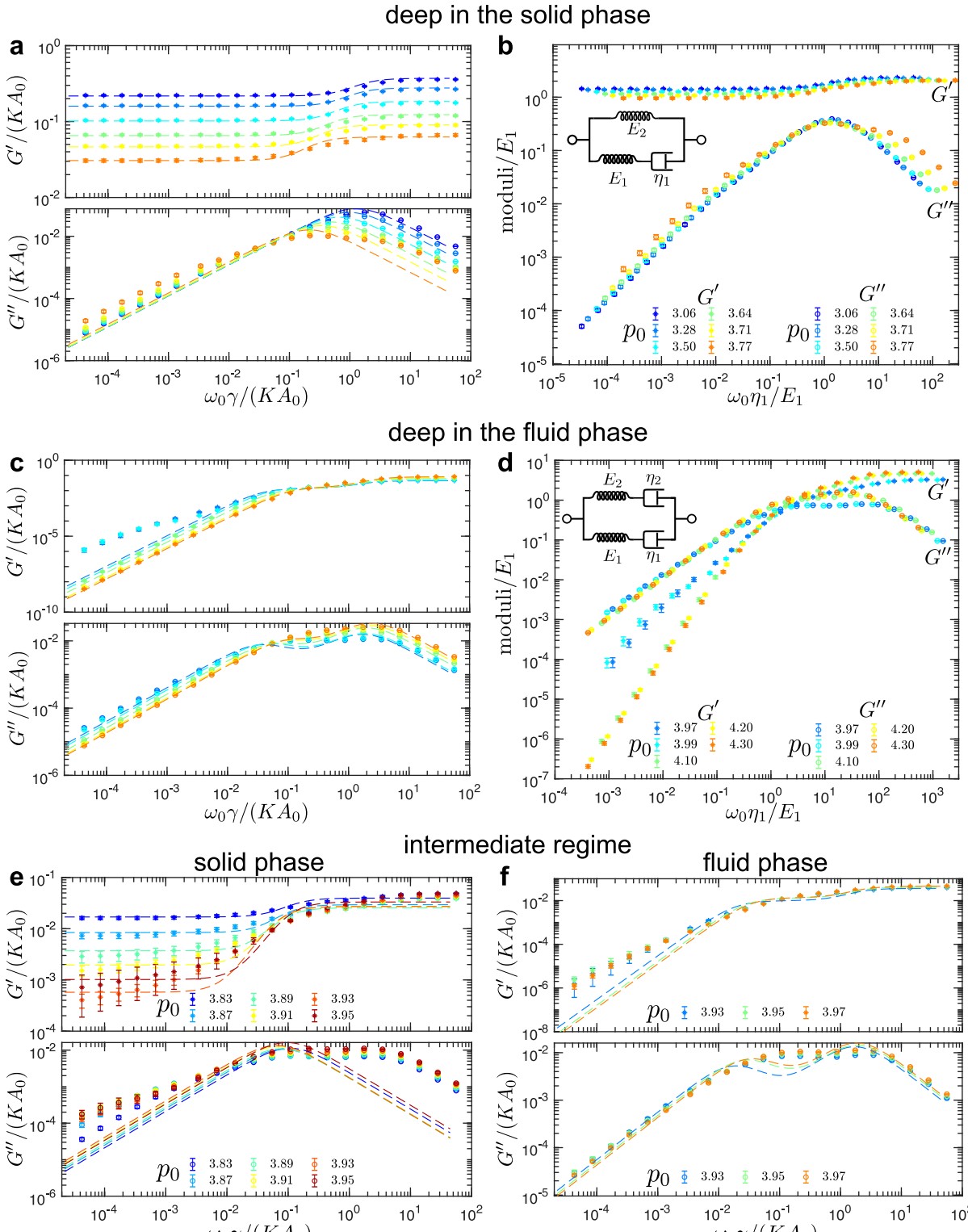

**Fig 4. Average storage and loss shear moduli in the solid and fluid phase for disordered tilings.** (a,c) Average storage ($G'$) and loss ($G''$) shear moduli as functions of the shearing frequency, $\omega_0$, for different values of the cell-shape parameter, $p_0$, (a) deep in the solid phase and (c) deep in the fluid phase. The error bars represent the standard error of the mean. (b,d) The collapse of the moduli curves for different values of $p_0$ for (b) the solid phase and (d) the fluid phase. The insets show the representation of (b) the Standard Linear Solid (SLS) model and (d) the Burgers model in terms of the springs and dashpots. (e,f) Average storage ($G'$) and loss ($G''$) shear moduli as functions of the shearing

frequency, $\omega_0$, for intermediate values of the cell-shape parameter, $p_0$, in (e) the solid phase and (f) the fluid phase. Dashed curves are the fits based on (a,e) the SLS model in the solid phase [see Eq (5)] and (c,f) the Burgers model in the fluid phase [see Eq (8)]. A representative example of random cell configurations used to produce these plots is shown in Fig 1b.

stress $\sigma(t) = \frac{1}{2}[\hat{\sigma}_{xx}(t) + \hat{\sigma}_{yy}(t)]$. As in the simple shear test, we then computed the dynamic bulk modulus as $B^*(\omega_0) = \tilde{\sigma}(\omega_0)/\tilde{\epsilon}(\omega_0)$ from which we obtained the storage bulk modulus $B' = \text{Re}(B^*)$ and the loss bulk modulus $B'' = \text{Im}(B^*)$ (see Fig 6c and 6d).

In the solid phase, the storage bulk modulus is independent of the driving frequency and the loss bulk modulus is zero. This is because in the solid phase, the hexagonal tiling is stable to biaxial deformation and there is no relative motion of vertices with respect to the substrate, which is the sole source of dissipation. Thus the response of the system can be captured by a single spring $E_{\text{solid}}$ (Fig 6e, inset). The measured value of the storage bulk modulus matches the analytical prediction,

$$B_{\text{theory}} = 2KA_0 + \sqrt[4]{12}\Gamma p_0 \tag{9}$$

by Staple, *et al.* [50], where the hexagonal tiling is assumed to undergo affine deformation under biaxial deformation. Storage bulk moduli, normalized by $B_{\text{theory}}$, for different values of $p_0$ all collapse to 1 (Fig 6e).

In the fluid phase, the bulk response behavior of the system can be described by the SLS model (Fig 6f, inset). While it might appear counter-intuitive to model a fluid with the SLS model, this is a direct consequence of the fact that in the fluid state, the bulk modulus is finite but the shear modulus vanishes, i.e., the fluid flows in response to shear but resists bulk deformation. The fitted storage and loss bulk moduli for the SLS model [see Eq (5)] show an excellent match with the simulation data (Fig 6d). This was also confirmed in Fig 6f, where we collapsed the storage and loss bulk moduli for different values of $p_0$.

The fitted values of elastic spring and dashpot viscosity constants for different values of $p_0$ are plotted in Fig 7. In the fluid phase, the storage bulk modulus in the high frequency limit $B'$ $(\omega_0 \to \infty) = E_1 + E_2$ [see Eq (5)] continuously increases from the value for the solid phase

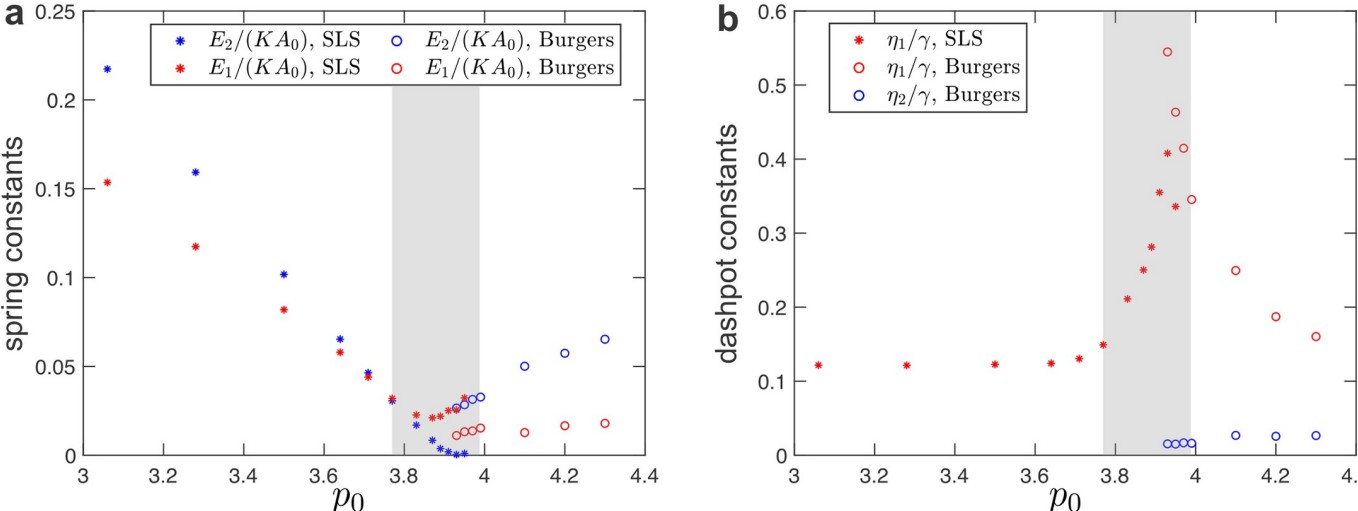

**Fig 5. Fitted values of spring-dashpot models for disordered tilings under simple shear.** (a) Elastic constants as a function of the target cell-shape parameter, $p_0$. (b) Dashpot viscosity constants as a function of the target cell-shape parameter, $p_0$. The shaded regions indicate the intermediate regime between the solid and fluid phases.

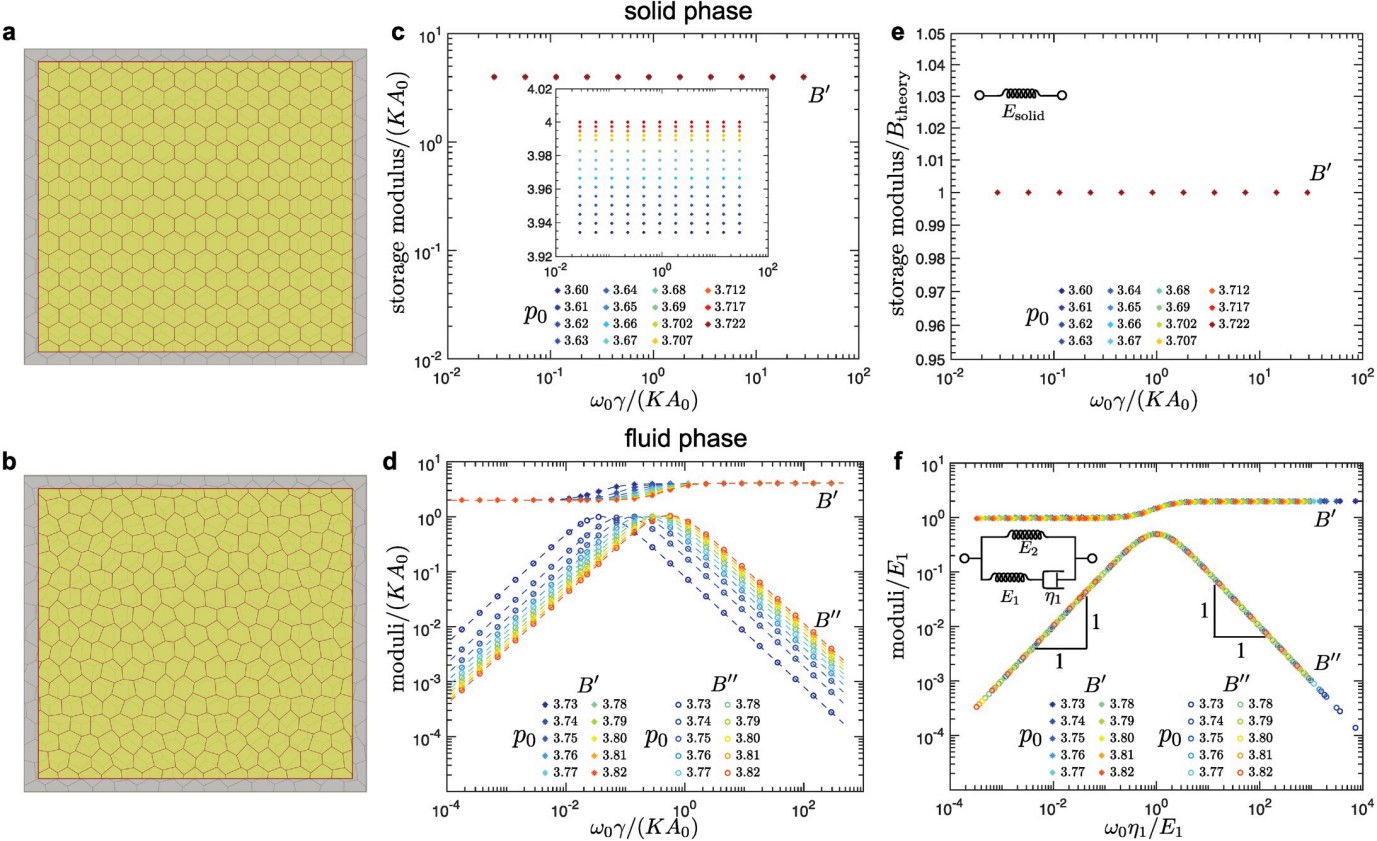

**Fig 6. Loss and storage bulk moduli in the solid (top row) and fluid phase (bottom row) for hexagonal tilings.** (a-b) An overlay of the representative reference (grey) and biaxially deformed (yellow) configurations in (a) the solid and (b) the fluid phase. The magnitude of the bulk deformation is highly exaggerated for demonstration purposes. (c-d) Representative storage ($B'$) and loss ($B''$) bulk moduli as functions of the deformation frequency, $\omega_0$, for different values of the cell-shape parameter, $p_0$. For the solid phase in (c), the loss bulk modulus $B'' \equiv 0$. For the fluid phase in (d), dashed curves are the fits based on the Standard Linear Solid (SLS) model [see Eq (5)]. (e-f) The collapse of the moduli curves for different values of $p_0$ for (e) the solid phase and (f) the fluid phase. The insets show the representation of (e) the spring model and (f) the SLS model in terms of the springs and dashpots. In panel (e), $B_{\mathrm{theory}}$ corresponds to the analytical prediction in Eq (9) for the storage bulk modulus in the solid phase.

$B_{\mathrm{theory}}$ [see Eq (9)] as the system transitions from solid to fluid (Fig 7a). The storage bulk modulus in the quasistatic limit $B'(\omega_0 \to 0) = E_2$ [see Eq (5)] emerges at the transition point with a finite value and increases as $p_0$ increases from $p_c$ (Fig 7a). Fig 7b shows that the dashpot constant $\eta_1$ diverges as the $p_0$ decreases toward $p_c$. Thus, the characteristic timescale $\eta_1/E_1$ also diverges (Fig 7c), but for a different reason than for the shear deformation, where the spring constant $E_1$ is vanishing (see Fig 3). Finally, we note that, unlike for the response to shear, the values of the spring and dashpot constants for bulk deformation are not sensitive to the local energy minimum configuration used to probe the response in the fluid phase, which is reflected by the very small errorbars in Fig 7. This is because the bulk moduli are dominated by the changes in cell areas.

The same procedures were applied to the ensemble of disordered tilings to probe the bulk rheology. Fig 8 shows the average storage and loss bulk moduli for different values of $p_0$. When the system is deep in the solid phase (Fig 8a) or deep in the fluid phase (Fig 8c), the bulk rheology can be described by the SLS model, which is confirmed by the fits (dashed curves) and the collapse in Fig 8b and 8d. The fitted values of spring and dashpot constants for different values of $p_0$ are shown in Fig 9. As $p_0$ approaches the value of solid-fluid transition, the fits based on

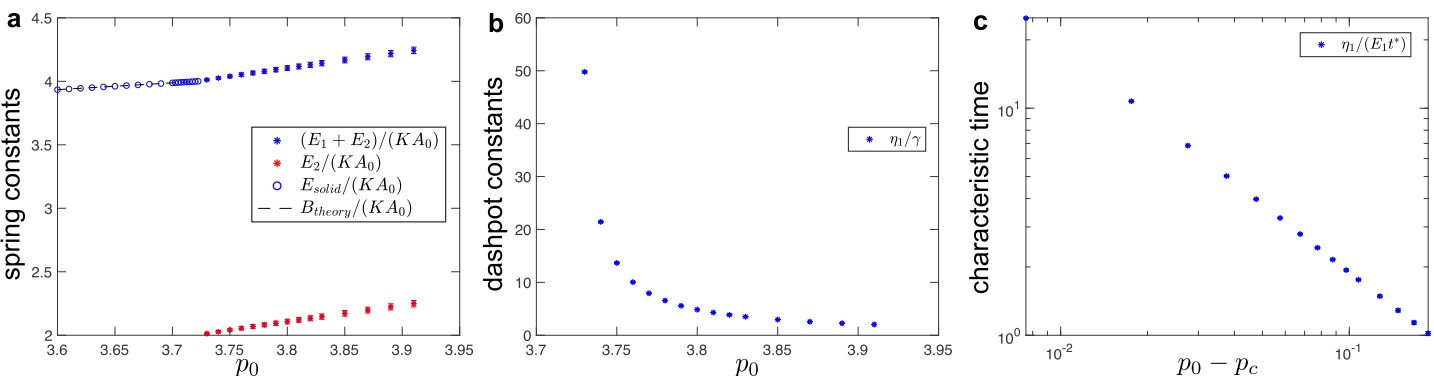

**Fig 7. Fitted values of spring-dashpot models for the system under bulk deformation as a function of the target cell-shape parameter, $p_0$.** (a) Elastic constants as a function of the target cell-shape parameter, $p_0$. In the solid phase ($p_0 < p_c \approx 3.722$), the bulk storage modulus $E_{\text{solid}}$ agrees with the analytical prediction $B_{\text{theory}}$ in Eq (9) (dashed line). At the solid-fluid transition point ($p_0 = p_c \approx 3.722$), it continuously changes to the high frequency limit of the bulk storage modulus, i.e., $B'(\omega_0 \to \infty) = E_1 + E_2$, of the fluid phase. The low frequency limit of the bulk storage modulus is $B'(\omega_0 \to 0) = E_2$ in the fluid phase. (b) Dashpot viscosity constant as a function of the target cell-shape parameter, $p_0$. (c) Characteristic timescales in the fluid phase obtained from the fitted values of the elastic constant and the dashpot viscosity. The normalization factor $t^* = \gamma/(KA_0)$ sets the unit of time. For the fluid phase ($p_0 > p_c \approx 3.722$), errorbars correspond to the standard deviation for simulations that were repeated for configurations that correspond to different local energy minima.

the SLS model deviate from the measured moduli curves (Fig 8a and 8c). At intermediate frequencies the storage moduli have a lower slope than predicted by the SLS model and the peak in the loss moduli is flattened and a second peaks starts to develop ($p_0 = 3.71$, $3.77$, and $3.80$ in Fig 8a and $p_0 = 3.99$ in Fig 8c). Fig 8e shows the storage and loss moduli for values of $p_0$ near the solid-fluid transition, and the collapsed data is shown in Fig 8f. As the value of $p_0$ approaches the critical value, the spread of the moduli increases, especially for low frequencies, which is seen in Fig J in the Sec J in S1 File, that presents the raw data of storage and loss bulk moduli at different values of $p_0$.

## Response to a shear deformation of a uniaxially pre-deformed system

The solid-fluid transition for the regular hexagonal tiling occurs when $p_0 \approx 3.722$, above which the hexagonal tiling is unstable. This is consistent with the vanishing of the affine shear modulus in Eq (7) at the transition point. If the regular hexagonal tiling is pre-compressed or pre-stretched uniaxially by a factor $a$, which is described by the deformation gradient $\hat{F} = \begin{pmatrix} a & 0 \\ 0 & 1 \end{pmatrix}$, then the high frequency limit of the linear storage shear modulus that is dominated by affine deformation becomes (see Sec G in S1 File),

$$G'_{\text{affine}}(a) = \frac{2\sqrt{2}\Gamma}{3^{7/4}a}\left(1 + \frac{1}{(1+3a^2)^{\frac{3}{2}}}\right)\left(-3p_0 + \sqrt[4]{192}(1 + \sqrt{1+3a^2})\right). \quad (10)$$

By setting the affine shear modulus to 0, we obtained the solid-fluid transition boundary in the $a - p_0$ plane as

$$p_c(a) = \sqrt{8\sqrt{3}}\,\frac{(1+\sqrt{1+3a^2})}{3}. \quad (11)$$

The above analytical prediction for the phase boundary (Fig 10a, blue line) shows an excellent agreement with the stability analysis in terms of the eigenvalues of the Hessian matrix $\frac{\partial^2 E}{\partial \mathbf{r}_i \partial \mathbf{r}_j}$ of the energy function [74] (Fig 10a, red dots). A given configuration is stable if all eigenvalues of the Hessian matrix are positive and the loss of mechanical stability occurs when the lowest

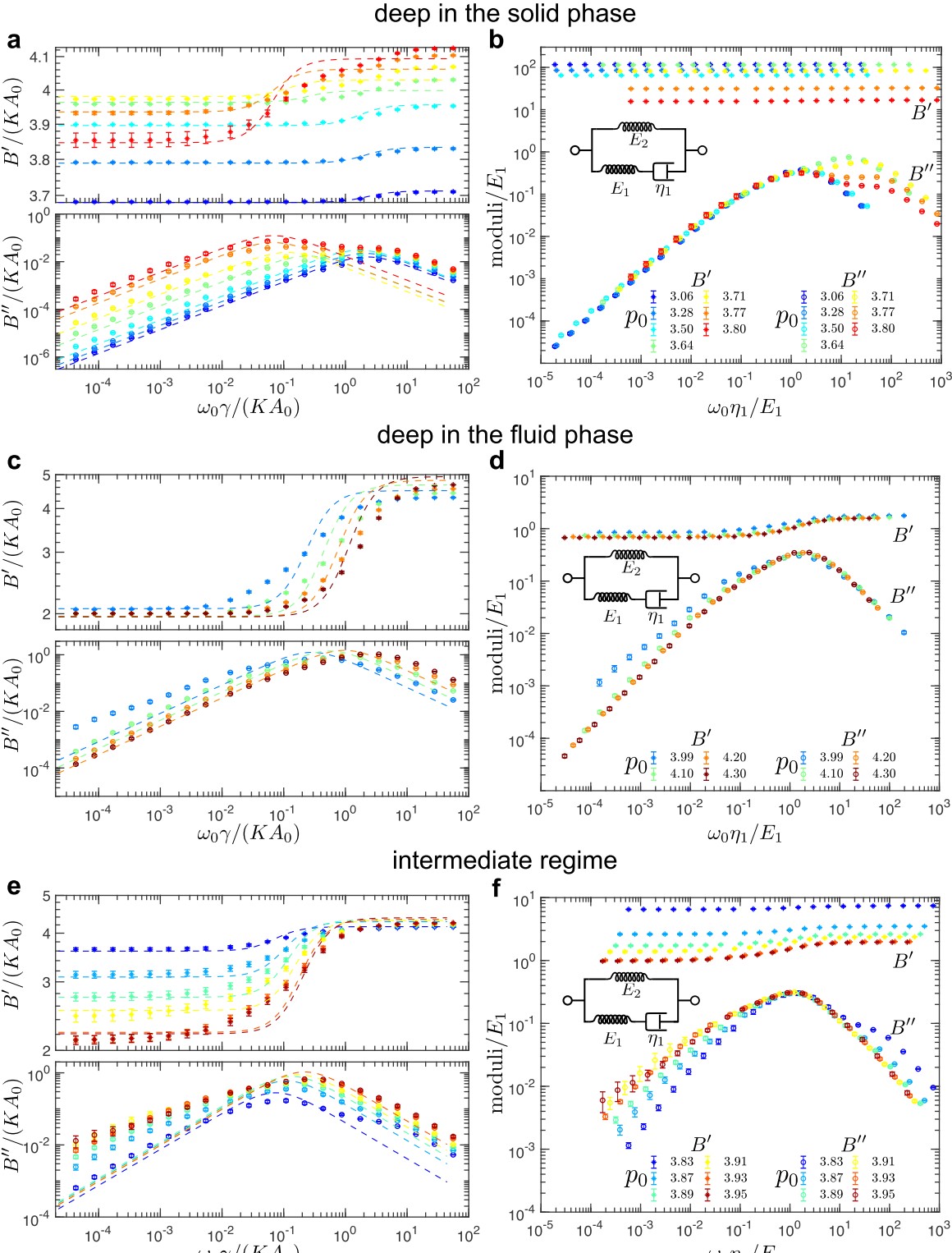

**Fig 8. Average storage and loss bulk moduli in the solid and fluid phase for disordered tilings.** (a,c) Average storage ($B'$) and loss ($B''$) bulk moduli as functions of the deformation frequency, $\omega_0$, for different values of the cell-shape parameter, $p_0$, (a) deep in the solid phase and (c) deep in the fluid phase. The error bars represent the standard error of the mean. (b,d) The collapse of the moduli curves for different values of $p_0$ for (b) the solid phase and (d) the fluid phase. The insets show the representation of the Standard Linear Solid (SLS) model in terms of the springs and dashpots. (e) Average storage ($B'$) and loss ($B''$) bulk moduli as functions of the deformation frequency, $\omega_0$, for intermediate values of the cell-shape parameter, $p_0$. (f) The collapse of the moduli curves for for intermediate values of the cell-shape parameter, $p_0$. Dashed curves in (a,c,e) are the fits based on the SLS model [see Eq (5)].

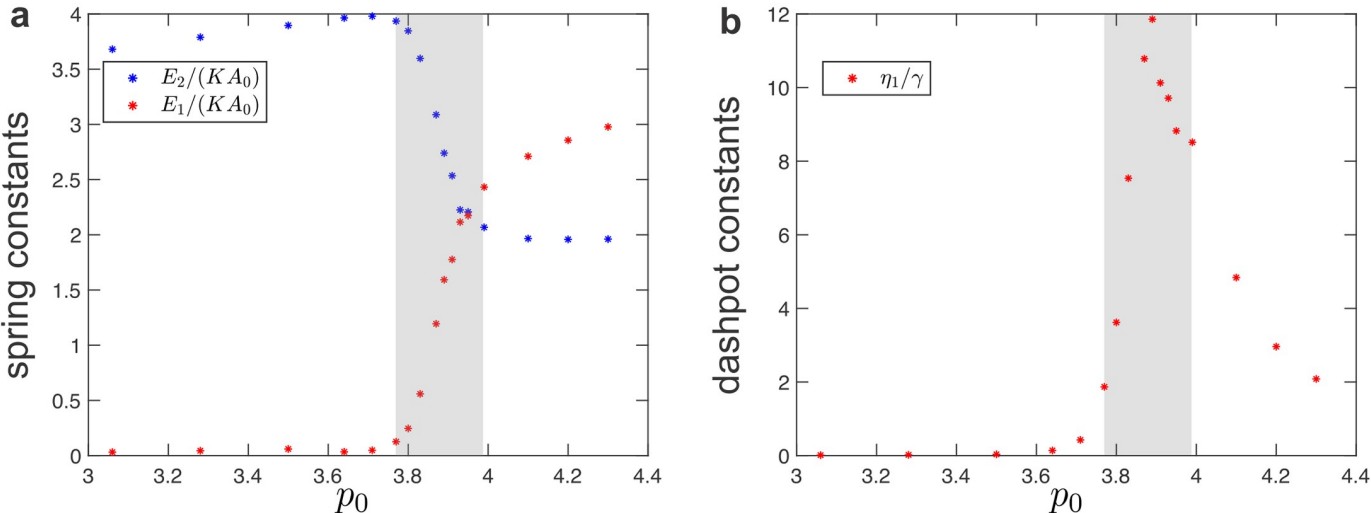

**Fig 9. Fitted values of spring-dashpot models for disordered tilings under bulk deformation.** (a) Elastic constants as a function of the target cell-shape parameter, $p_0$. (b) Dashpot viscosity constant as a function of the target cell-shape parameter, $p_0$. The shaded regions indicate the intermediate regime between the solid and fluid phases.

eigenvalue becomes 0. For a given $p_0$, the value of the lowest eigenvalue reduces with decreasing $a$, i.e., as the magnitude of compression is increased. Thus, the compression (stretching) shifts the solid-fluid transition towards the lower (higher) values of $p_0$ (see Fig 10a).

We also probed the response to oscillatory shear applied to uniaxially pre-compressed and pre-stretched systems. This analysis was done on the uniaxially deformed hexagonal tiling in the solid phase as well as a system in the fluid phase obtained by relaxing the unstable, uniaxially deformed hexagonal tiling after an initial random perturbation (see Model and methods). The response to the shear deformation is qualitatively similar and can still be described by the SLS model in the solid phase and the Burgers model in the fluid phase. Fig 10b and 10c shows fitted values of the parameters for spring-dashpot models when the system is under uniaxial compression ($a = 0.95$), no pre-deformation ($a = 1.00$, i.e., same as Fig 3a and 3b), and uniaxial

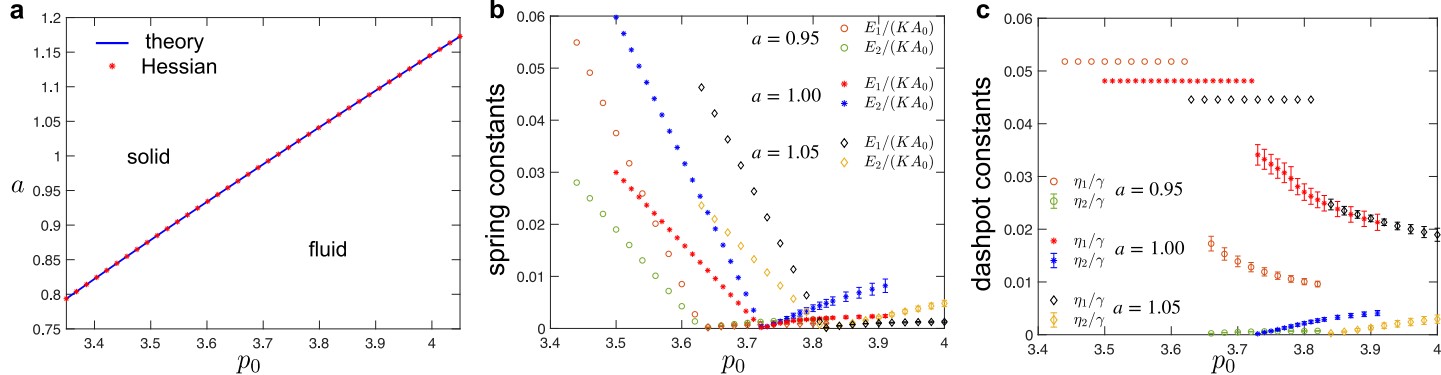

**Fig 10. Tuning the solid to fluid transition by applying uniaxial pre-deformation.** (a) The solid-fluid transition boundary in the $a - p_0$ plane, where $a$ measures the amount of uniaxial pre-deformation described by the deformation gradient $\hat{F} = \begin{pmatrix} a & 0 \\ 0 & 1 \end{pmatrix}$. Blue line shows the analytical prediction from Eq (11), which matches the stability analysis with the Hessian matrix (red dots). (b,c) The fitted values of the (b) spring and (c) dashpot constants for the SLS model in the solid phase [see Eq (5)] and the Burgers model in the fluid phase [see Eq (8)] when the system is under uniaxial compression ($a = 0.95$), no pre-deformation ($a = 1.00$), and under uniaxial tension ($a = 1.05$).

tension ($a = 1.05$). In both the solid and fluid phases, all spring elastic constants decrease to 0 as $p_0$ approaches the critical value predicted by Eq (11). The dashpot constant $\eta_1$ remains constant in the solid phase. Once the system enters the fluid phase as $p_0$ increases, a new dashpot constant $\eta_2$ emerges and increases from 0, while the value of the dashpot constant $\eta_1$ decreases. As in the simple shear case, we note that the dashpot constant $\eta_1$ has a discontinuous jump at the solid-fluid transition (see Fig 3c) and that the values of the spring and dashpot constants are somewhat sensitive to the local energy minimum configuration used to probe the response in the fluid phase. The errorbars in Fig 3 show standard deviation for configurations that were obtained by using different random initial perturbation (see Model and methods). Finally, we note that besides the uniaxial pre-deformation, the solid-fluid transition point can be tuned by other modes of pre-deformation (see Fig G and Sec G in S1 File).

## Discussion and conclusions

We have performed a detailed analysis of the rheological properties of the vertex model subject to small-amplitude oscillatory deformations over seven orders of magnitude in the driving frequency. Our analysis shows that the vertex model exhibits non-trivial viscoelastic behavior that can be tuned by a single dimensionless geometric parameter—the shape parameter, $p_0$. In order to characterize the response, we constructed constitutive rheological models that use combinations of linear springs and dashpots connected in series and in parallel. These models allowed us to match the shear response of the vertex model to that of the Standard Linear Solid model in the solid phase and the Burgers model in the fluid phase. In the low-frequency, i.e., quasistatic regime, our results are fully consistent with many previous studies [20, 21, 50, 72]. Our work, however, provides insights into the time-dependent response of the vertex model over a broad range of driving frequencies, which is important if one is to develop full understanding of the rheological properties of the vertex model and how they inform our understanding of epithelial tissue rheology.

While the SLS and the Burgers model accurately describe rheology of the vertex model of disordered tilings deep in the solid and liquid phases, respectively, these models deviate from the data for $p_0$ values in the vicinity of the solid-fluid transition. This is because close to the transition points additional relevant time scales start to emerge. As shown in Fig K in the Sec K in S1 File, adding additional Maxwell elements in parallel to the spring-dashpot models increases accuracy of the fits. This is to be expected since each Maxwell element introduces a new time scale. The physical interpretation of these additional time scales has clearly to do with the local arrangements of the cells for a particular disordered configuration but tying it to a specific cell pattern is, however, not easy. In addition, we also found that for disordered tilings the loss shear modulus crosses over from the linear scaling in frequency at low $\omega_0$ to the $\sim \omega^\alpha$ with $\alpha \approx 0.73$ at intermediate frequencies (Fig L in the Sec L in S1 File). The crossover moves to lower frequencies as the system size increases. This behavior suggests a large (potentially infinite) number of relevant timescales.

It is important to note that we considered only friction between cells and the substrate and neglected any internal dissipation within the tissue. Therefore, the dissipation is solely due to relative motion of the cells with respect to the substrate as a result of non-affine relaxation of the tissue. The approach used in this study, therefore, would not be suitable for modeling the rheological response of epithelia not supported by a solid substrate, e.g., for early stage embryos or suspended epithelia in the experiments of Harris, et al. [37]. Furthermore, dissipative processes in epithelia are far more complex than simple viscous friction and are not fully understood. It has, for example, recently been argued that internal viscoelastic remodelling of the cortex can lead to interesting collective tissue behaviors [75]. We have addressed some of

these questions in a separate recent work [76] using a semi-analytic approach based on the normal-mode expansion.

In this work we kept the ratio $\Gamma/(KA_0)$ fixed, since this ratio does not qualitatively change the behavior of the vertex model [20, 41]. In Ref. [76], however, we further explored the effect of parameter $\Gamma/(KA_0)$, where we showed that the perimeter stiffness $\Gamma$ and the area stiffness $K$ affect the low-frequency and high-frequency rheological behavior, respectively.

We also showed that the critical value for the solid-fluid transition can be tuned by applying pre-deformation. Interestingly, under uniaxial and biaxial (i.e., isotropic) pre-compression the solid to fluid transition shifts to lower values of $p_0$, leading to the non-intuitive prediction that one can fluidize the system by compressing it. This is, however, unsurprising, since the transition is driven by a geometric parameter that is inversely proportional to the square root of the cell's native area. Compressing the system reduces its area and, hence, effectively increases $p_0$. It is, however, important to note that this is just a property of the vertex model and it does not necessarily imply that actual epithelial tissue would behave in the same way. Cells are able to adjust their mechanical properties in response to applied stresses, and it would be overly simplistic to assume that compression would directly lead to changes in the preferred area. In fact, experiments on human bronchial epithelial cells show that applying apical-to-basal compression, which effectively expands the tissue laterally (i.e., corresponds to stretching in our model), fluidizes the tissue [19].

Furthermore, the transition from solid phase to fluid phase is accompanied by the emergence of a large number of soft modes. As we have noted, it has recently been argued that these soft modes lead to a nonlinear response distinct from that obtained in classical models of elasticity [45]. Approximately half of the eigenmodes are zero modes (see Fig H in the Sec H in S1 File). While the analysis of soft modes in the vertex model is an interesting problem [53], it is beyond the scope of this work. Other models in this class have intriguing non-trivial mechanical properties, such as the existence of topologically protected modes [77–81].

We briefly comment on the values of vertex model parameters and timescales that are relevant for experimental systems. While obtaining accurate in vivo measurements of elastic coefficients of epithelial tissues is notoriously difficult, it is possible to make order of magnitude estimates. For example, recent experiments on human corneal epithelial cells estimated $K/\gamma \approx 0.5 \ \mu\text{m}^{-2}\text{h}^{-1}$ and $A_0 \approx 500 \ \mu\text{m}^2$ [25]. This would correspond to the timescale $\gamma/(KA_0) \approx$ 15 s, or the relevant frequencies in the $\sim 10^0$–$10^2$ Hz range. Characteristic values of stress have been estimated to be $KA_0 \sim 10\text{nN}/\mu\text{m}$ for a number of different epithelia [71, 82].

It is also important to note that our work focuses on the rheological behavior in the tangent moduli approximation to a general stress-strain curves, where applied shear and bulk deformations are infinitesimal with respect to a potentially significantly predeformed state (e.g., see Fig 10). This is clearly an idealized case and, in reality, one would be interested in the response to higher values of the applied deformation ($\gtrsim 1 - 10\%$). It would be possible to study such finite deformations using our simulation protocol. Interpreting the results would, however, be more challenging, e.g., due to shear-driven rigidity transition in model tissues and the presence of plastic events [44, 46].

The vertex model provides a rather coarse description of real epithelial tissues and omits many important aspects such as cell polarization, chemical signalling, cells' ability to actively adjust their properties in response to their environment, etc. The framework presented in this study would, however, be able to address the linear response in the presence of such effects, provided that the vertex model is suitably augmented [83–85].

Regardless of whether cells in an epithelial tissue are arrested or able to move, the rheological response of the tissue is viscoelastic with multiple timescales [38]. This response arises as a result of the complex material properties of individual cells combined with four basic cellular

behaviors: movement, shape change, division, and differentiation. The tissue not only has a non-trivial rheological response but is also able to tune it. There is growing evidence that this ability of biological systems to tune their rheology, and in particular, transition between solid-like and fluid-like behaviors, plays a key role during morphogenesis [4]. How such cellular processes are regulated and coordinated to form complex morphological structures is only partly understood. It is, however, clear that the process involves mechano-chemical feedback between mechanical stresses and the expression of genes that control the force-generating molecular machinery in the cell. Any models that aim to describe morphological processes, therefore, need to include coupling between biochemical processes and mechanical responses. The base mechanical model, however, must be able to capture the underlying viscoelastic nature of tissues. Our work provides evidence that the vertex model, a model commonly used to study the mechanics of epithelial tissues, has interesting non-trivial rheological behavior. This, combined with its ability to capture both fluid- and solid-like behavior by tuning a single geometric parameter shows it to be an excellent base model to build more complex descriptions of real tissues.

## Supporting information

**S1 File. Supporting information file.**
(PDF)

## Acknowledgments

We would like to acknowledge useful discussions with Ricard Alert, Moumita Das, and Mikko Haataja.

## Author Contributions

**Conceptualization:** Sijie Tong, Rastko Sknepnek, Andrej Košmrlj.

**Data curation:** Sijie Tong, Navreeta K. Singh.

**Formal analysis:** Sijie Tong, Navreeta K. Singh, Rastko Sknepnek, Andrej Košmrlj.

**Funding acquisition:** Rastko Sknepnek, Andrej Košmrlj.

**Investigation:** Sijie Tong, Navreeta K. Singh, Rastko Sknepnek, Andrej Košmrlj.

**Methodology:** Sijie Tong, Rastko Sknepnek, Andrej Košmrlj.

**Project administration:** Rastko Sknepnek, Andrej Košmrlj.

**Resources:** Rastko Sknepnek, Andrej Košmrlj.

**Software:** Sijie Tong, Rastko Sknepnek.

**Supervision:** Rastko Sknepnek, Andrej Košmrlj.

**Validation:** Rastko Sknepnek, Andrej Košmrlj.

**Visualization:** Sijie Tong, Navreeta K. Singh.

**Writing – original draft:** Sijie Tong, Rastko Sknepnek, Andrej Košmrlj.

**Writing – review & editing:** Sijie Tong, Navreeta K. Singh, Rastko Sknepnek, Andrej Košmrlj.

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
