## [Decision Letter · Decision Letter 0]

28 Feb 2022

Dear Dr. Kosmrlj,

Thank you very much for submitting your manuscript "Linear viscoelastic properties of the vertex model for epithelial tissues" for consideration at PLOS Computational Biology. As with all papers reviewed by the journal, your manuscript was reviewed by members of the editorial board and by several independent reviewers. The reviewers appreciated the attention to an important topic. Based on the reviews, we are likely to accept this manuscript for publication, providing that you modify the manuscript according to the review recommendations.

Both reviewers are quite positive and support publication of this work but they highlight several specific issues that appear constructive and should help avoid several questions other readers would likely have. If you decide to submit a revised version, please address these remaining reviewers' concerns. Please also make sure that all code underlying your findings is made available.

Sincerely,

Tobias Bollenbach

Associate Editor

PLOS Computational Biology

Jason Haugh

Deputy Editor

PLOS Computational Biology

[LINK]

Both reviewers are quite positive and support publication of this work but they highlight several specific issues that appear constructive and should help avoid several questions other readers would likely have. If you decide to submit a revised version, please address these remaining reviewers' concerns. Please also make sure that all code underlying your findings is made available.

Reviewer's Responses to Questions

**Comments to the Authors:**

Reviewer #1: The manuscript “Linear viscoelastic properties of the vertex model for epithelial tissues” by Tong et al. explores the rheological properties of epithelial tissue in computer simulations using the vertex model. In contrast to previous models which were considering mechanical properties of the vertex model on long time scales only, this analysis now provides insight into viscoelastic mechanical properties at a large range of time scales.

The analysis is presented for ordered and disordered hexagonal tilings of the epithelium in the solid-like and fluid-like regime as tuned by the shape parameter p_0=P_0/√(A_0 ). Here, P_0 is the preferred cellular perimeter and A_0 is the preferred cellular surface area. Furthermore, the authors report the rheological properties for disordered polygonal tilings at different values of the parameter p_0. For both cases, regular hexagonal and irregular polygonal tilings, moduli with regards to shear and bulk deformations are reported. Interestingly, the authors find that for most parameter regimes, the mechanics of the epithelium can be captured by established viscoelastic models such as the Standard Linear Solid (SLS) and the Burgers model, which can describe the full time-scale dependent mechanics of the material with three or four material constants (two elastic constants and one or two viscosities), respectively. The authors analyse how these effective material constants depend on the shape parameter p_0, the nature of the deformation (shear or bulk) and the initial configuration of the epithelium (regular hexagonal or irregular polygonal). Within their approach the authors consider only friction between cells and the substrate and neglect internal dissipation within the tissue.

My impression is that the research has been carried out with high technical rigor and that the paper is well written and thought through. Furthermore, I can see that the work makes a substantial new contribution to the field.

Therefore, I recommend to publish the paper in PLOS Computational Biology after revision with regards to the few queries that I listed below.

Major:

The authors state that deformations have been applied in their study at a small amplitude of ϵ_0=10^(-7) to ensure that the mechanical response is in the linear regime. Clearly, that order of magnitude of deformation is not of any practical relevance in reality. Therefore, can the authors clarify, to which extent their analysis changes if deformations are stepped up to higher values such as the 1-10% range?

Minor:

Figure 4 would benefit from an additional panel where an exemplary irregular polygonal lattice is shown.

Section 7 of the Supplement would profit from a more detailed explanation or referencing to account for the derived equations.

Reviewer #2: The manuscript by Tong. et al. carefully explores the viscoelastic properties of the Vertex model- a basic model that has been extensively used to study biophysical properties of epithelial tissues. The authors analyze the response of the cellular network under small shear and bulk deformations for both a regular hexagonal network and irregular topological configurations. They found that standard spring-dashpot models quantitatively explain the complex behaviors of the Vertex model under these deformations. I found the method, analysis, and results fascinating and recommend this paper for publication. However, I recommend the authors elaborate on a few details:

1- It is not clear how the periodic boundary conditions are implemented: First, how do they treat the PBC to ensure no external stress on cells? How is the PBC implemented for simple shear simulations? The correct implementation of PBC for simple shear simulations requires careful movement of the vertices in each image boundary.

2- According to [50], the ground state transition from solid hexagonal network to liquid irregular network happens in multiple steps: First transition line occurs at a state that cells are free to change their shape, but the topology should be hexagonal. After the second transition line, cells can freely change their shapes, but only 5, 6, and 7 sided cells are allowed, and so on. Consequently, the shear modulus vanishes in multiple transitions, and a fully disordered network will have zero shear modulus only for p<p_0. about="" authors="" have="" in="" seen="" simulations="" the="" their="" these="" thought="" transitions="">

3- Cell size variation dramatically increases for a larger value of \\Gamma_C. Have the authors done simulations to measure tissue rheology for different values of \\Gamma_C?

4- Cells in many tissues have anisotropic mechanical properties. For example, in a fly embryo or wing, cells are polarized, and cellular polarizations are mostly aligned (all cells have the same polarization orientation). Proteins that determine the polarization of cells also affect cell mechanics such that some boundaries are under higher tension than others. Can the authors comment on how tissue polarization might affect tissue rheology?</p_0.>

**Have the authors made all data and (if applicable) computational code underlying the findings in their manuscript fully available?**

Reviewer #1: **No: **The authors state that the simulation and analysis codes and initial configurations are available upon request.

Reviewer #2: Yes

PLOS authors have the option to publish the peer review history of their article (what does this mean?). If published, this will include your full peer review and any attached files.

Reviewer #1: No

Reviewer #2: No

Figure Files:

Data Requirements:

Reproducibility:

References:

---

## [Editor Report · Decision Letter 1]

25 Apr 2022

Dear Dr. Kosmrlj,

We are pleased to inform you that your manuscript 'Linear viscoelastic properties of the vertex model for epithelial tissues' has been provisionally accepted for publication in PLOS Computational Biology.

Best regards,

Tobias Bollenbach

Associate Editor

PLOS Computational Biology

Jason Haugh

Deputy Editor

PLOS Computational Biology

---

## [Editor Report · Acceptance letter]

12 May 2022

PCOMPBIOL-D-22-00136R1

Linear viscoelastic properties of the vertex model for epithelial tissues

Dear Dr Košmrlj,

I am pleased to inform you that your manuscript has been formally accepted for publication in PLOS Computational Biology. Your manuscript is now with our production department and you will be notified of the publication date in due course.

With kind regards,

Livia Horvath
